# 18F-FDG PET/CT Cannot Substitute Endoscopy in the Staging of Gastrointestinal Involvement in Mantle Cell Lymphoma. A Retrospective Multi-Center Cohort Analysis

**DOI:** 10.3390/jpm11020123

**Published:** 2021-02-13

**Authors:** Tetiana Skrypets, Cristina Ferrari, Luca Nassi, Gloria Margiotta Casaluci, Benedetta Puccini, Lara Mannelli, Kateryna Filonenko, Irina Kryachok, Felice Clemente, Maria Carmela Vegliante, Antonella Daniele, Gianmauro Sacchetti, Attilio Guarini, Carla Minoia

**Affiliations:** 1Haematology Unit, IRCCS Istituto Tumori “Giovanni Paolo II”, 70124 Bari, Italy; tetianaskrypets@gmail.com (T.S.); felice.clemente.irccsbari@gmail.com (F.C.); mariella.vegliante@live.it (M.C.V.); attilioguarini@oncologico.bari.it (A.G.); 2D.I.M.—Diagnostic Imaging-Nuclear Medicine, University of Bari “Aldo Moro”, 70124 Bari, Italy; ferrari_cristina@inwind.it; 3Haematology Department, Azienda Ospedaliero-Universitaria Maggiore della Carità, 28100 Novara, Italy; luca.nassi@med.uniupo.it (L.N.); gloria.margiotta@med.uniupo.it (G.M.C.); 4Hematology Department, Azienda Ospedaliera Careggi, 50139 Firenze, Italy; benedettapuccini@virgilio.it (B.P.); la.mannelli25@gmail.com (L.M.); 5Oncohematology Department, National Cancer Institute, 03022 Kyiv, Ukraine; ksfilonenko@yahoo.com (K.F.); irina.kryachok@gmail.com (I.K.); 6Experimental Oncology and Biobank Management Unit, IRCCS Istituto Tumori “Giovanni Paolo II”, 70124 Bari, Italy; anto.dani27@gmail.com; 7Nuclear Medicine, AOU Maggiore della Carità, 28100 Novara, Italy; gianmauro.sacchetti@maggioreosp.novara.it

**Keywords:** Mantle Cell Lymphoma, staging, endoscopy, gastric biopsy, colorectal biopsy, 18F-FDG PET/CT

## Abstract

The detection of gastrointestinal (GI) involvement in Mantle Cell Lymphoma is often underestimated and may have an impact on outcome and clinical management. We aimed to evaluate whether baseline 18F-FDG PET/CT presents comparable results to endoscopic biopsy in the diagnosis of GI localizations. In our retrospective cohort of 79 patients, sensitivity and specificity of 18F-FDG PET/CT were low for the stomach, with a fair concordance (*k* = 0.32), while higher concordance with pathologic results (*k* = 0.65) was detected in the colorectal tract. Thus, gastric biopsy remains helpful in the staging of MCL despite 18F-FDG PET/CT, while colonoscopy could be omitted in asymptomatic patients. The validation of our data in prospective cohorts is desirable

## 1. Introduction

Mantle Cell Lymphoma (MCL) is a rare aggressive B-cell non-Hodgkin lymphoma (B-NHL) [1]. It represents 3% of all NHL, with male predominance and a median age at onset of about 60 years. Patients usually have an advanced stage disease, with almost 70% of them presenting a stage IV for bone marrow (BM) or other extranodal involvement [2]. It presents an unfavourable clinical course when treated with conventional chemotherapy, and a high rate of recurrence [2]. The reciprocal translocation t(11;14) resulting in the overexpression of cyclin D1 is pathognomonic for MCL, constituting a pathogenetic chromosomal abnormality and also a target for diagnostics tests specific for the disease [1]. Diagnosis is established by histological or extra-nodal tissue together with basic immunohistochemistry panel, including CD5+, CD10−/+, CD20+, CD23−/+, CD43+, cyclin D1+ [1].

The correct staging of the disease is fundamental in order to define the therapeutic iter. Initial stage disease (stage I/II) can be treated with chemo-immunotherapy combined with radiotherapy. On the contrary, for advances stage disease the use of high-dose aracytin alternated with chemo-immunotherapy and consolidation with stem cell transplant (ASCT) is recommended for young and fit patients [3]. Current staging of the disease is carried out with computed tomography (CT), fluorine-18-fluorodeoxyglucose positron emission tomography/computed tomography (18F-FDG PET/CT) and BM biopsy, and could include gastrointestinal (GI) endoscopy, topic of the present study [4,5].

Current guidelines recommend 18F-FDG PET/CT to stage FDG-avid lymphomas, including MCL [4,5]. This technique showed a high accuracy rate in the detection of both nodal (sensitivity 100%) and extranodal involvement of lymphoma [6]. The principal sites of extranodal localization in MCL patients are BM and the GI tract. However many other tissues could be involved: rhinopharynx, orbit, lung, central nervous system, etc. Although 18F-FDG PET/CT has replaced the use of BM biopsy, an invasive exam, for the staging of patients diagnosed with classical Hodgkin Lymphoma, as recommend by the current guidelines [4,5,6,7], in the field of B-NHL, BM biopsy cannot actually be spared to the patient [4,5]. In particular, despite its high specificity (98%), recent data showed that 18F-FDG PET/CT cannot replace BM biopsy in the staging of MCL patients because of low sensitivity (52%), while the sensitivity of this technique for other extranodal sites has not been well established yet [8]. 

GI involvement was historically reported in about 15–30% of patients diagnosed with MCL. Higher percentages (74%) have been reported within prospective trials, including 43% in the upper and 88% in the lower GI tract [9,10]. GI involvements were mostly asymptomatic and affected multiple segments. The most affected GI tracts were stomach and colon, which could result normal or present some endoscopic abnormalities, mainly erythema, thickness, nodularity, ulceration or micropolyposis. In the case of endoscopic abnormalities, the biopsy confirmed the presence of MCL, but an infiltration was also found in the two thirds of normal mucosa or in non-specific endoscopic findings [9,10]. The higher percentages of infiltration were found when a random biopsy had been performed for each anatomic site (stomach, duodenum, ascending, transverse, and descending colon) [9]. Additional molecular analysis (fluorescence in situ hybridization (FISH) for t(11;14) and polymerase chain reaction (PCR) for immunoglobulin heavy chain gene) could be performed in the histologic specimen, thus biopsy constitutes the unique tool to confirm MCL infiltration [10,11]. 

18F-FDG PET/CT remains the best non-invasive exam able to stage MCL patients, but it demonstrated a low sensitivity for GI involvement, as reported in the recent study by Albano et al. [8]. The majority of the available studies focused on the GI localizations evaluated by endoscopic exams. Lee et al. demonstrated worse prognostic features among patients with GI involvements, who presented an advanced stage disease, a worse clinical prognostic score and a higher rate of recurrence of the disease. Recurrences were also diagnosed using endoscopic exams, thus confirming the central role of endoscopy for the detection of GI localizations of MCL [12].

Nowadays, we do not have clear data whether 18F-FDG PET/CT is able to detect GI involvement and eventually guide or substitute endoscopic exams. The bottom line seems to be that gastric lesions could be of benign or inflammatory nature, and the exact diagnosis can be obtained only after biopsy. 

We have therefore conducted a multi-center retrospective study, with the aim to (i) evaluate the accuracy of 18F-FDG PET/CT in the detection of GI involvement compared to biopsy, and (ii) establish whether it is possible to omit endoscopy in this subset of patients, thus accelerating the staging period and sparing invasive tests. 

## 2. Materials and Methods

### 2.1. Study Design

We retrospectively included in the study consecutive patients diagnosed with MCL from October 2006 to October 2020. The Institutions involved in the study were: Hematology Unit, IRCCS Istituto Tumori “Giovanni Paolo II”, Bari, Italy (coordinating center); Haematology Department, Azienda Ospedaliero-Universitaria Maggiore della Carità, Novara, Italy; Haematology, University of Firenze, Firenze, Italy; Department of Oncohematology, National Cancer Institute, Kyiv, Ukraine.

Inclusion criteria were: age > 18 years, histologically proven diagnosis of MCL (World Health Organization 2016) [1] anddisease staging through CT scan, 18F-FDG PET/CT scan and BM biopsy for all patients. In addition, according to the clinical local practice, patients performed esophagogastroduodenoscopy (EGD)/colonoscopy with biopsy at staging. After induction chemotherapy, the response to treatment was established by post-treatment 18F-FDG PET/CT scan. EGD/colonoscopy were repeated after post-induction therapy only if positive at baseline. BM biopsy was also repeated if positive at staging, according to guidelines. Cheson’s revised criteria for malignant lymphoma were used to assess the stage and response to treatment [5]. Patient’s characteristics were collected from health records and reported in a dedicated database.

Data collected at the time of diagnosis included: age, sex, diagnosis according to WHO classification (including histological variants), Ki67% in the lymph node, data of diagnosis, stage, MCL International Prognostic Index (MIPI) and simplified MIPI, BM biopsy infiltration, presence of extra-nodal disease and site of extra-nodal disease, 18F-FDG PET/CT positivities on the GI tract and its description, EGD findings and their description, gastric biopsy result, colonoscopy findings and their description, colorectal biopsy and description. Patients started chemotherapy immediately after the baseline examinations, with different regimens according to age and eligibility for high-dose chemotherapy. Data on induction chemotherapy were also collected: type of chemotherapy, data of start and conclusion, number of courses, consolidation with ASCT, maintenance therapy. Response to therapy was assessed: final 18F-FDG PET/CT, EGD and gastric biopsy findings, colonoscopy and biopsy findings. Data of recurrence was also collected. 

The protocol was approved by each local ethics committee, in accordance with the principles of the Declaration of Helsinki and the Guidelines for Good Clinical Practice, and patients signed informed consent. 

### 2.2. 18F-FDG PET/CT Imaging Protocol and Interpretation

Patient preparation included fasting for at least 6 h, adequate hydration, blood glucose measurement, and an intravenous injection of 18F-FDG whose dose depended on the tomograph and the patient’s weight, with subsequent uptake time of 45–60 min. In all centers, patients were instructed to void before imaging acquisition, and no oral or intravenous contrast agents were administrated or bowel preparation used for any patient. The PET/CT scanner was a Discovery PET/CT system (Bari: GE Healthcare, Milwaukee, WI, USA; Novara and Firenze: Siemens Biograph 16 Hi Rez, Knoxville, TN, USA; Kyiv: Siemens Biograph 64, Munich, Germany). Supine decubitus under the scanner with arms in front of the pelvis is the position chosen for optimal image reading and interpretation and to avoid overlapping the arms with the spine. The fields of view include from the skull to the mid-thigh (5–7 bed positions). The Discovery system includes a multidetector helical CT scanner. We evaluated gastric and colorectal positivities. Every radiotracer uptake deviating from physiologic distribution and background was regarded as suggestive of lymphoma.

Both focal and diffuse GI 18F-FDG uptakes were considered for the analysis.

### 2.3. Endoscopy

Patients performed EGD and/or colonoscopy with biopsy at staging according to local practice. A description of endoscopic findings was reported and included: normal mucosa/negative, gastritis, duodenitis, micropolyposis, ileitis, colitis. A biopsy was collected in all cases. In patients with normal mucosa/aspecific findings, a random biopsy has been obtained. Histological exam with confirmed MCL localization included immunoistochemical analyses performed with the following markers: CD5, CD10, CD20, CD23, CD43, cyclin D1. The presence of ciclin D1 was obtained either with immunohistochemistry or with FISH analysis for t(11;14).

### 2.4. Statistical Analysis

A descriptive analysis was performed for sample description and frequencies. The overall sensitivity (Se), specificity (Sp), positive predictive value (PPV), negative predictive value (NPV) and accuracy (Ac) of 18F-FDG PET/CT for gastric and intestinal involvement detection were calculated by using biopsy results as gold standard (95% confidence interval). MedCalc Software (MedCalc Software Ltd., Ostend, Belgium) for Windows was used for statistics. Cohen’s K test was performed using R package “psych” within “R” statistical software v 3.6.1 and used to estimate the degree of agreement between 18F-FDG PET/CT and endoscopy plus biopsy performances. Result interpretation had been defined as “slight agreement”—0.01–0.20; “fair agreement”—0.21–0.40; “moderate agreement”—0.41–0.60; “substantial agreement”—0.61–0.80; “almost perfect or perfect agreement”—0.81–1.00 [13].

## 3. Results

### 3.1. Patients’ Characteristics

We retrospectively evaluated 79 patients for the analysis (6.3% presenting a blastoid variant). The median age was 66.8 years (range, 27–83). The majority of patients (*n* = 71, 89.9%) presented an advanced stage disease (III/IV), of whom 58 (73.4%) had BM infiltration. MCL International Prognostic Index (MIPI) was prevalently intermediate (*n* = 28, 35.5%) or high (*n* = 31, 39.2%). 5 patients presented a blastoid histologic variant (6.3%). Patients’ characteristics are detailed in Table 1.

18F-FDG PET/CT was able to identify 9 (11.4%) extranodal localizations of MCL other than BM and GI. Orbital (22.2%) and liver (22.2%) involvements were the most frequent extranodal sites of disease. Other extra-nodal localizations were: rinopharynx, soft palate, lung, pleura and adrenal gland.

The majority of patients eligible for intensified therapy had received R-CHOP regimen (rituximab, cyclophosphamide, doxorubicin, vincristine, prednisone) alternated to R-DHAP/Ox (rituximab, dexamethasone, high-dose aracytin, cisplatin/ xaliplatin) (*n* = 16, 20.2%), or R-CHOP and high-dose aracytin (*n* = 3, 3.8%), or R-BAC500 (rituximab, bendamustine, aracytin) (*n* = 4, 5.1%). Twenty-one patients (26.5%) underwent ASCT.

Patients not eligible for intensified chemotherapy received R-CHOP (*n* = 17, 21.5%), R-BAC500 (*n* = 10, 12.7%), R-bendamustine (*n* = 22, 27.8%) or R-CVP (rituximab, cyclophosphamide, prednisone) (*n* = 3, 3.8%). Eleven patients (13.9%) proceeded with rituximab maintenance. Finally, 4 patients (5.1%) followed an initial “watch and wait” approach.

The overall response rate to first line chemotherapy was 90.7% with a complete response for most patients (*n* = 60, 80%). Relapses/progressions were observed in 52.3% of patients.

### 3.2. GI Endoscopic and Histopathologic Findings

EGD had been performed in 52 (65.8%) patients. The following endoscopic findings were reported: normal mucosa/negative (*n* = 34), gastritis (*n* = 15), micropolyposis (*n* = 2), duodenitis (*n* = 1). Thirteen gastric biopsies were found positive for MCL localization (25%). Chronic gastritis (1 Helicobacter pylori positive) was histologically documented in 4 cases (7.7%).

Colonoscopy had been performed in 31 (39.2%) patients. Endoscopic findings were: normal mucosa/negative (*n* = 21), micropolyposis (*n* = 7), colitis (*n* = 2), ileitis (*n* = 1). Eleven colorectal biopsies were positive for MCL localization (35.5%).

In all biopsy samples, the histological confirmation of MCL involvement included the positivity for ciclin D1 either with immunohistochemistry or with FISH analysis.

### 3.3. Correlation between PET/CT Results and Endoscopic Biopsies

GI positivity on 18F-FDG PET/CT was found in 30 patients (38%). Gastric uptake was present in 18 cases (22.8%) and was mainly described as diffuse in the gastric wall. Intestinal abnormal 18F-FDG uptake was found in 12 cases (15.2%) either as diffuse (*n* = 9) or focal (*n* = 3) in the right iliac fossa or as colorectal accumulation.

In Figure 1 is described a representative case from our cohort.

Compared with EGD/gastric biopsy results (*n* = 52), the two methods were discordant in 15/52 (28.8%) patients for upper GI involvement. According to Cohen’s k test, 18F-FDG PET/CT and EGD/gastric biopsy showed a fair agreement (*k* = 0.32, 71.2%). The false-negative rate of 18F-FDG PET/CT was 9.6% (*n* = 5), while the false-positive rate was 19.2% (*n* = 10). Most false-positivities were described as gastritis and nonspecific lesions at histology. The Se of 18F-FDG PET/CT in the upper GI tract was 61.54% and Sp 74.36%. Ac, PPV and NPV were 71.15%, 44.44% and 85.29% respectively.

Compared with colonoscopy/colorectal biopsy (*n* = 31), we observed that the two methods were discordant in detecting colorectal involvement in 5/31 (16.1%) patients. Agreement between 18F-FDG PET/CT and colonoscopy/colorectal biopsy was substantial (*k* = 0.65, 83.9%). The false-negative rate of 18F-FDG PET/CT was 6.5% (*n* = 2) while the false-positive rate was 9.7% (*n* = 3). Se was 81.82% and Sp 85%. Ac, PPV and NPV were 83.87%, 75% and 89.47% respectively. All results are summarized in Table 2.

## 4. Discussion

Accurate staging presents a primary impact on the outcome of MCL patients. 18F-FDG PET/CT shows high performances in many avid-lymphoma histotypes and also in MCL, in which it has demonstrated good accuracy in the detection of nodal involvement with the highest specificity [6,14,15,16]. However, its accuracy in the definition of BM and GI involvement seems to be suboptimal [4,8]. For GI evaluation with 18F-PET/CT there are, though, only few articles in the literature [14,15,16]. Thus, the value of 18F-FDG PET/CT in this field is still unclear and its potential utility is not completely established.

Considering that GI involvement could occur up to 80–90% of patients, a correct definition of this aspect at diagnosis presents important clinical implications in the staging, in the choice of the induction chemotherapy and to assess the final response to therapy and monitoring [17]. In fact, induction chemotherapy differs between stage I-II disease and advanced stage disease (including that with GI infiltration) [3,18]. Moreover, patients with GI involvement could present specific symptoms (pain, bleeding, anaemia) and then benefit from a strengthened supportive care.

Despite this high prevalence, current international guidelines do not consider as mandatory the execution of endoscopic procedures for all patients. In fact, they recommend EGD and colonoscopy in symptomatic patients and for those who present limited stages at 18F-FDGPET/CT [3,18]. In this context, we also have to consider that often there is a need to rapid beginning of anti-neoplastic treatment due to symptoms and impairing performance status and not all patients result eligible for GI endoscopy. For this reason, there is the need of imaging methods that can represent non-invasive markers of disease activity.

In the present study, we found results concerning Se, quite comparable to that of Albano et al. However, we found that 18F-FDG PET/CT performance resulted to be still lower than endoscopy plus biopsy, especially for the upper GI tract. In particular, in our analysis, 18F-FDG PET/CT resulted positive at the GI level in 38% of patients (gastric 22.8%, intestinal 15.2%). We assessed that Se and Sp of 18F-FDG PET/CT compared to histology were low for the stomach (61.54% and 74.36%, respectively), mainly due to the presence of false positives caused by gastritis or nonspecific lesions, with a fair concordance (K 0.32, 71.2% of agreement). For the detection of colorectal localizations, in our cohort 18F-FDG PET/CT demonstrated higher concordance with pathologic results (k 0.65, 83.9% of agreement) with almost 90% of NPV.

The available literature cohorts on the topic confirmed the role of the endoscopic exams during the staging and surveillance of MCL patients [8,12,19,20]. Additionally, patients presenting a GI localization seemed to have worse prognostic score and higher rate of recurrence [12]. Our study specifically focused on the performances of the 18F-FDG PET/CT exam in the detection of GI localizations of MCL and on the possibility of substituting invasive endoscopic exams in the staging, and then in the monitoring of the disease. Available reports described performances of 18F-FDG PET/CT evaluated overall in the GI tract, without conclusive information on the upper and lower tract [8,19], thus not being comparable with our results. Albano et al. found out a Se, Sp, PPV, NPV, and accuracy of 18F-FDG PET/CT for GI in general of 64%, 91%, 69%, 90%, and 85%, concluding for a suboptimal sensitivity of the exam [8].

However, our analysis firstly demonstrated the performances of 18F-FDG PET/CT distinctly in the detection of gastric and intestinal localization of MCL compared to endoscopic findings and allowed to propose a practical management of these patients.Even with the limitations of the number of included patients and the rate of positive biopsies, the studyledto the conclusion that gastric biopsy remains helpful in the staging of MCL despite 18F-FDG PET/CT, which presented low Se and Sp in this tract. Conversely, colonoscopy, which is surely invasive and debilitating, could be omitted in asymptomatic patients, according to the high rate of agreement between endoscopy and 18F-FDG PET/CT.

## 5. Conclusions

Even if to date 18F-FDG PET/CT cannot substitute endoscopy in the staging of gastrointestinal involvement in MCL, it remains helpful in the staging of MCL as whole-body non-invasive routinely used imaging method. Our results demonstrate that 18F-FDG PET/CT could replace colonoscopy in asymptomatic patients, leading to optimization of management and therapeutic approach, thus avoiding useless procedures in a significant percentage of patients, while EDG should be offered to patients even if asymptomatic.

This study could be of practical help in the clinical management of MCL patients, allowing an integration to previous reports and guidelines. The validation of our data in larger cohorts and in prospective clinical trials is desirable.

## Figures and Tables

**Figure 1 jpm-11-00123-f001:**
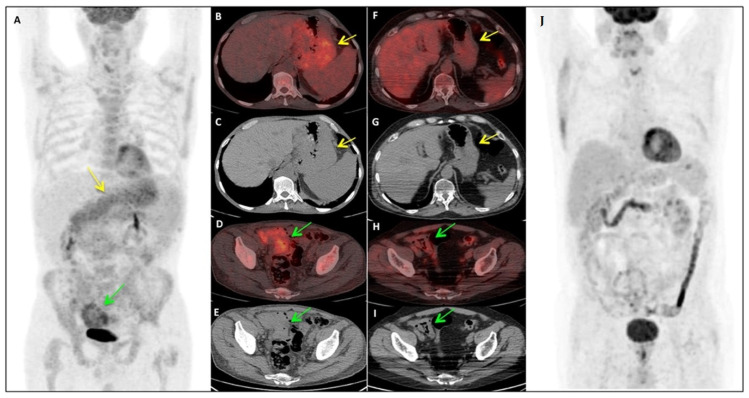
Pre- and post-therapy 18F-FDG PET/CT assessment in a patient affected by MCL. The patientis a 63-years old man diagnosed with a classic stage IV MCL. He presented an important gastric and intestinal involvement at staging (**A**–**E**). After treatment with R-CHOP/R-DHAP, we could observe a normalization of the 18F-FDG PET/CT findings (**F–J**). In this patient, also gastric and colorectal biopsy were found positive. (**A**–**E**) Baseline Maximum Intensity Projection (MIP)(**A**), axial fused PET/CT (**B**,**D**) and CT images (**C**,**E**): PET/CT demonstrated pathological 18FDG uptake (SUV max 4.9) in the whole stomach wall (**B**, yellow arrow), which appear thickened at the reference co-registered CT images (**C**, yellow arrow), as well as in the intestinal walls (SUV max 5.2) in the right iliac fossa (**D**, green arrow), which appear conglomerate (**E**, green arrow). (**F**–**J**) Post-therapy evaluation MIP (**J**), axial fused PET/CT (**F**,**H**) and CT images (**G**,**I**): PET/CT demonstrated complete disappearance of 18F-FDG uptake both in stomach and intestinal walls (**F**,**H**), which returned to be of normal thickening also at the co-registered CT images (**G**,**I**).

**Table 1 jpm-11-00123-t001:** Patients’ baseline characteristics.

Patients’ Characteristics	*N* = 79	−100%
Age at evaluation, years (mean, range)	66.8 (27–83)	
Male	52	65.8
Female	27	34.2
Stage	79	100
I/II	8	10.1
III/IV	71	89.9
B-symptoms	79	100
Present	8	10.1
Absent	71	89.9
BM infiltration	79	100
Yes	58	73.4
No	21	26.6
Extra-nodal involvement (other than BM and GI)	79	100
Yes	9	11.4
No	70	88.6
Site of extra-nodal involvement (other than BM and GI) by PET/CT	9	11.4
orbit	2	22.2
liver	2	22.2
rinopharynx	1	11.1
soft palate	1	11.1
lung	1	11.1
pleura	1	11.1
adrenal gland	1	11.1
MIPI score	79	100
Low risk (0–5.7)	20	25.3
Intermediate risk (5.7–6.2)	28	35.5
High risk (>6.2)	31	39.2
Simplified MIPI score	79	100
Low risk (0–3)	19	24.1
Intermediate risk (4–5)	40	50.6
High risk (>6)	20	25.3
Ki 67%	48	60.7
≤30%	25	52
>30%	23	48
Blastoid variant	79	100
Yes	5	6.3
No	74	93.7
Type of induction chemotherapy	79	100
R-CHOP/R-DHAP or R-DHAOx	16	20.2
R-CHOP/HD-AraC	3	3.8
R-CHOP	17	21.5
R-BAC500/HD-AraC	4	5.1
R-BAC500	10	12.7
R-B	22	27.8
R-CVP	3	3.8
Watch and wait	4	5.1
Autologous stem cell transplant	21	26.5
Rituximab maintenance	11	13.9
Response to induction chemotherapy	75	94.9
CR	60	80
PR	8	10.7
SD	0	0
PD	7	9.3
Disease relapse/progression	65	82.3
Yes	34	52.3
No	31	47.7

BM, bone marrow; GI, gastrointestinal; MIPI, Mantle Cell Lymphoma International Prognostic Index; R-CHOP, rituximab, cyclophosphamide, doxorubicin, vincristine, prednisone; R-DHAP, rituximab, dexamethasone, high-dose aracytin, cisplatin; R-DHAOx, rituximab, dexamethasone, high-dose aracytin, oxaliplatin; R-BAC, rituximab, bendamustine, aracytin; R-B, rituximab, bendamustine; R-CVP, rituximab, cyclophosphamide, prednisone; CR, complete response; PR, partial response; SD, stable disease; PD, progressive disease.

**Table 2 jpm-11-00123-t002:** Performance of 18F-FDG PET/CT in detecting GI involvement by MCL, compared with endoscopy and biopsy.

18F-FDG PET/CT (*n* = 79, 100%)	EGD/Gastric Biopsy (*n* = 52, 65.8%)	Colonoscopy/Colorectal Biopsy (*n* = 31, 39.2%)
Se (95%CI)	61.54% (31.58–86.14%)	81.82% (48.22–97.72%)
Sp (95%CI)	74.36% (57.87–86.96%)	85% (62.11–96.79%)
PPV (95%CI)	44.44% (28.72–61.36%)	75% (50.47–89.83%)
NPV (95%CI)	85.29% (74–92.20%)	89.47% (70.54–96.79%)
Accuracy (95%CI)	71.15% (56.92–82.87%)	83.87% (66.27–94.55%)
Cohen’s k test (95%CI)	0.32, agreement 71.2%	0.65, agreement 83.9%

Se, sensitivity; Sp, specificity; PPV, positive predictive value; NPV, negative predictive value.

## Data Availability

The data presented in this study are available on request from the corresponding author. The data are not publicly available due to ethical.

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
