# Peer review of "18F-FDG PET/CT Cannot Substitute Endoscopy in the Staging of Gastrointestinal Involvement in Mantle Cell Lymphoma. A Retrospective Multi-Center Cohort Analysis"

_jpm, 2021, doi:10.3390/jpm11020123_

Round 1

Reviewer 1 Report

The authors conducted a retrospective multicenter study including 79 mantle cell lymphoma patients, and demonstrated that FDG-PET/CT can not substitute gastroscopy and biopsy, but it can substitute colonoscopy to diagnose GI involvement.

As an endoscopist, I would like to know the most efficient endoscopic diagnosis for GI involvement in MCL. What are the endoscopic findings suggestive of MCL involvement? Should endoscopists perform biopsy even if endoscopic finding is normal? What part of the GI tract is involved most frequently in MCL? What is the optimal number of biopsy to identify GI involvement in MCL?  Please reply these questions from this study or literature, and also please describe issues below in the manuscript.    

  1. Detail of endoscopic finding (normal, gastritis, duodenitis, ileitis, colitis, tumor, polyp, etc.).
  2. Did endoscopists performed biopsy only when there were findings suggestive of lymphoma, or did they do biopsy even if endoscopic finding was normal?
  3. Please demonstrate detail of biopsy (site, number)
  4. Reason to perform or not to perform endoscopy (e.g., uptake in PET/CT, symptomatic (pain, GI bleeding), screening)

In addition, I would like to know how GI involvement diagnosis influences the management of MCL. Does it change chemotherapy protocol?

Table 2 is difficult to comprehend. Showing both percentage and fraction (e.g. 23% (12/52)) seems better.  

Author Response

Authors thanks the Reviewer for comments, which ameliorate the quality of the manuscript.

Please find enclosed the point-by-point responses to all requests. 

Reviewer 2 Report

In this paper, the authors investigated if 18F-FDG PET/CT would present sufficient information to replace/omit endoscopic biopsy in the detection of gastrointestinal involvement in Mantle Cell Lymphoma. I find this paper to be well-written, well-organized, easy to read. The motivation and methodology were clearly explained. The findings were clearly presented and would be of interest to the journal's readers.

My only concern is that the cohort of 79 patients does not truly represent the analyses as there were only 52 EGD and 31 colonoscopy patients. Moreover, the positive biopsy rates in EGD/colonoscopy were low to make a conclusive statement. The authors should consider adding such limitations to the Discussion.

Minor comments:

Line 36: Mantle Cell Lymphoma >> Mantle Cell Lymphoma (MCL)

Line 60: BM was defined earlier.

Line 76: invasive misspelled.

Line 107: Please add company details i.e., company name, city, country for MedCalc Software

Line 118: missing word/verb in “of whom 58 (73.4%) a BM infiltration”

Lines 200 and 208: No need for a new paragraph.

Line 220: Patients >> Patient

Line 222: RCHOP >> R-CHOP and RDHAP >> R-DHAP (for consistency)

Line 224: MIP was not defined.

Line 279: was low >> were low

Line 285: available misspelled.

Author Response

Authors sincerely thanks the Reviewer for the positive comments to our manuscript. They also thank the Reviewer for the suggestions, which ameliorate the quality of this manuscript.

Please find enclosed the point-by-point-response. 

Round 2

Reviewer 1 Report

The revised manuscript has been improved, and it is worth to publish. 

Reviewer 2 Report

No further comments.